# Effects of Jujube Powder on Growth Performance, Blood Biochemical Indices, and Intestinal Microbiota of Broiler

**DOI:** 10.3390/ani13213398

**Published:** 2023-11-02

**Authors:** Jing Liang, Zejian Nie, Yapeng Zhao, Shizhen Qin, Fang Nian, Defu Tang

**Affiliations:** 1College of Animal Science and Technology, Gansu Agricultural University, Lanzhou 730070, China; liangjing0320@outlook.com (J.L.); 18298411081@163.com (Z.N.); 17693491365@163.com (Y.Z.); qinshizhen@163.com (S.Q.); 2College of Science, Gansu Agricultural University, Lanzhou 730070, China; nianf@gsau.edu.cn

**Keywords:** broiler, jujube powder, growth performance, physical and chemical indices of blood, intestinal microbiota

## Abstract

**Simple Summary:**

Broiler feeding now faces great challenges because of the rising cost of poultry diets, with corn being the main component. Developing and utilizing unconventional feed resources is an effective method of feeding broilers, thereby alleviating the problem of a high-cost diet. This study investigated how supplementation with jujube powder influences the growth performance, nutrient apparent utilization, serum immune indices, antioxidant indices, and intestinal microbiota of Cobb broilers. Adding jujube powder to the diet significantly improved the average daily gain, the apparent utilization rate of organic matter, and the apparent metabolic energy of broilers; it enhanced the body’s immunity and antioxidant performance and improved the intestinal microbiota of broilers. The most appropriate percentage of jujube powder to be supplemented was 8%. This indicates that jujube powder can act as a new feed type to replace some basic broiler diets.

**Abstract:**

In total, 576 Cobb broilers were randomized into 6 treatment groups, with 8 replicates in each treatment group and 12 broilers in each replicate. Each treatment group was fed six different experimental diets containing 0%, 2%, 4%, 6%, 8%, and 10% jujube powder. The group receiving 0% jujube powder was considered the blank control group. The experimental period was 42 days and was divided into two periods: starter (0–21 days) and finisher (22–42 days). Compared with the control group, the addition of 8% jujube powder significantly improved the ADG of broilers (*p* < 0.05), and 8% and 10% jujube powder significantly improved the total tract apparent digestibility of organic matter in broilers (*p* < 0.05). Adding 10% jujube powder significantly improved the apparent metabolic energy of broilers (*p* < 0.05). Compared with the control group, 4–10% jujube powder significantly increased IgA, IgG, IgM, and sCD4 levels (*p* < 0.05) and T-AOC and SOD contents, and it reduced the MDA content in the serum of broilers (*p* < 0.05). In addition, the relative abundance of Firmicutes, Bacteroidetes, *Lactobacillus*, and *Romboutsia* significantly increased in the broiler ileum, whereas that of Proteobacteria and *Enterobacter* decreased significantly (*p* < 0.05) when 8% jujube powder was added to the diet. The relative abundance of Proteobacteria, *Bacteroides,* and *Faecalibacterium* in the cecum increased significantly (*p* < 0.05), whereas that of Bacteroidetes decreased significantly (*p* < 0.05).

## 1. Introduction

Chickens are the most common domesticated animal. Compared with other types of meat, chicken meat is rich in proteins, trace elements, carnosine, creatinine, and amino acids [1,2]. Thus, broiler feeding is irreplaceable. However, the cost of poultry feed, with corn being the main component, has recently been increasing. To promote the sustainable development of the poultry industry and effectively alleviate the problem of high feed costs, developing and utilizing unconventional feed resources has become an inevitable new trend [3,4,5,6].

Red jujube is abundant in China [7], and a large amount of poor-quality jujube is discarded as agricultural waste. Small, scarred, and defective jujubes can be used to prepare jujube powder as an unconventional feed for animals, thereby achieving a win–win situation for fruit farmers and poultry producers. The basic nutrient content of jujube is similar to that of corn [8,9]. Thus, jujube can be used as a high-energy feed, replacing parts of corn in corn–soybean-based animal diets [10]. Furthermore, jujube is rich in minerals, vitamins, and various active substances, such as polysaccharides, phenols, terpenes, nucleosides, dietary fiber, and rutin, which exert dual regulatory effects of nutrition and physiology [11,12,13,14,15]. A diet supplemented with jujube powder can significantly improve the antioxidant capacity of serum, improve body metabolism, and enhance the immunity of laying hens [16]. A diet supplemented with yeast selenium and jujube powder can also maintain the structural stability of the chicken myofibrillar protein after slaughter, improve its digestion characteristics, and inhibit protein oxidation [17,18]. Jujube powder added to the diet can also enhance the growth performance, nutrient digestibility, and meat quality of goats [19]. Jujube also exerts inhibitory effects on cancer cells at different growth stages [20], and its bioactive components (ursolic acid and oleanolic acid) have antiproliferative and apoptotic effects on cancer cells (ovarian cancer cells and normal human fibroblasts BJ1-hTERT and nonmalignant breast epithelial MCF-10A cells) [21,22]. Jujube powder is currently known to mainly improve the meat quality and blood biochemistry in broilers, and the action mechanism of this powder on intestinal microorganisms in broilers is rarely reported. Therefore, in this experiment, defective jujubes were processed into a powder and added to the broiler diet to evaluate the effects of the addition of different amounts of jujube powder on the growth performance, nutrient utilization, blood physical and chemical indices, and intestinal microbiota of broilers. This information would provide a theoretical basis for the rational application of jujube powder in poultry production.

## 2. Materials and Methods

### 2.1. Test Materials

The jujube used in this experiment was collected from the jujube garden in Minqin County, Wuwei City, Gansu Province. The dehiscent jujube, ground jujube, and small jujube were selected manually. Moth-eaten and moldy jujubes were removed. The selected jujubes were then sun-dried in a well-ventilated place. The jujube powder was obtained by freezing the jujubes and then crushing the frozen jujubes using a freeze crusher.

The contents of nutrients in jujube powder were determined using conventional methods. The crude protein content was determined by the Kjeldahl method (GB/T 6432-2018 [23]), the crude fat content was determined by the Soxhlet extraction method (GB/T 6433-2006 [24]), the crude fiber content was determined by the filtration method (GB/T 6434-2006 [25]), the crude ash content was determined by the high-temperature burning method (GB/T 6438-2007 [26]), the calcium content was determined by the potassium permanganate titration method (GB/T 6436-2018 [27]), and the phosphorus content was determined by the spectrophotometric method (GB/T 6437-2018 [28]). The results of the measurements are presented in Table 1.

### 2.2. Test Design

In total, 576 1-day-old male Cobb broilers were randomized into 6 treatment groups, with 8 replicates in each group and 12 chickens in each replicate. They were fed corn–soybean meal-based diets supplemented with 0%, 2%, 4%, 6%, 8%, or 10% jujube powder. The test diet was formulated by referring to the agricultural industry standard of the People’s Republic of China chicken feeding standard (NY-T33-2004 [29]). Table 2 presents the composition and nutritional level of diets. The diets used in the present study were calculated to meet or exceed the National Research Council (NRC) nutrient requirements for broilers [30]. The experiment lasted for 42 days, which was divided into two stages: starter (0–21 days) and finisher (22–42 days). Because of winter feeding, the animal house was warmed up 2 days before the chickens entered the houses. The temperature was controlled using a water-heating boiler. It was maintained at 35 °C to 38 °C for 3 days and gradually cooled down to 20 °C at a rate of 1 °C every day. Then, the same temperature was maintained until the end of the experiment. Within 2 h after the experimental chicks arrived, low-concentration (5%) glucose aqueous solution was used for drinking, and the water temperature was moderate. In total, 12 chickens were reared in three-layer cages (length: 140 cm, width: 70 cm, and height: 40 cm), with 8 cages in each group. During the experimental period, the chickens were allowed free access to food and drink and exposed to light for 24 h. The relative humidity of the animal house was 50%. The animal house was ventilated normally, and the environment was kept clean. The animals were immunized in strict accordance with the Cobb broilers’ immunization management manual.

### 2.3. Observations

#### 2.3.1. Growth Performance

The broilers were weighed by pens at 0, 21, and 42 days of age. The amount of feed consumed by each pen was recorded over the starter and finisher periods. Mortality was recorded daily. Any bird that died was weighed, and the weight was used to adjust the feed/gain ratio (F/G). The F/G was calculated by dividing the total feed intake by the weight of live plus dead birds. The average daily feed intake (ADFI), average daily gain (ADG), and F/G were calculated as follows:(1)$$\;\mathrm{ADFI}=\frac{\mathrm{total}\;\mathrm{feed}\;\mathrm{intake}}{experimental\;days}$$
(2)$$\;\mathrm{ADG}=\frac{\mathrm{final}\;\mathrm{BW}\;-\;\mathrm{initial}\;\mathrm{BW}}{experimental\;days}$$
(3)$$\;F/G=\frac{\;\mathrm{feed}\;\mathrm{intake}}{\begin{array}{c} weight\;gain \end{array}}$$

#### 2.3.2. Coefficient of Total Tract Apparent Nutrient Digestibility or Retention

On the 35th day of the feeding experiment, three broilers in good body condition and of similar body weights were randomly selected from each group. They were fed in a single cage for the complete fecal metabolism test. On the 37th day, fasting was started at 20:00 to eliminate the effect of the intestinal contents of the broilers on the metabolism test. Free drinking water was made available during fasting, with no changes in other feeding conditions. At 08:00 of the 38th to 41st day, fresh excreta (removing debris, feed, and other debris) were collected for 4 consecutive days. Then, 10 mL of 10% hydrochloric acid was immediately added per 100 g of excreta, and the mixture was immediately refrigerated at −20 °C. Later, the frozen excrement was collected, thawed, mixed evenly, and dried at 65 °C to a constant weight. The excreta were weighed after being maintained at room temperature for 24 h to regain moisture. Then, the excreta were crushed through a 40-mesh sieve, added to a bag, and sealed for the subsequent test and analysis. The contents of total tract apparent dry matter and organic matter digestibility and total tract nitrogen retention in the excreta were determined by referring to the AOAC method [31]. Energy was measured using the oxygen and nitrogen calorimeter (ikc2000, IKA, Staufen, Germany). Titanium dioxide was determined by referring to the methods of Short et al. [32]. The total tract apparent nutrient digestibility or retention was calculated as follows:(4)$$\mathrm{Apparent}\;\mathrm{nutrient}\;\mathrm{digestibility}\;\mathrm{or}\;\mathrm{retention}=\frac{\left(Nutrient\;intake-Nutrient\;content\;in\;excreta\right)}{\begin{array}{c} \begin{array}{c} Nutrient\;intake \end{array} \end{array}}\times 100$$

#### 2.3.3. Serum Immune and Antioxidant Indices

At the end of the experimental period, 8 chickens (one for each repetition) were randomly selected from each treatment group. Blood was withdrawn from the chicken wing vein. After the blood sample was centrifuged at 4000 r/min for 15 min, the serum was separated, packed in a 2 mL centrifuge tube, and stored at −20 °C to determine immune and antioxidant indices. The serum immunoglobulin IgA, IgG, and IgM and the soluble CD4 surface antigen (sCD4) were determined using the ELISA kit (mlbio, Shanghai, China). The serum glutathione peroxidase (GSH-Px), malondialdehyde (MDA), superoxide dismutase (SOD), and total antioxidant capacity (T-AOC) were determined using the kit of Nanjing Jiancheng Biotechnology Co., Ltd., Nanjing, China.

#### 2.3.4. Intestinal Chyme Sample Collection and Microbial Sequencing

Six broilers with the best growth performance on day 42 were randomly selected from the control and treatment groups. After slaughtering these broilers, the ileum and cecum were stripped. The midgut chyme was collected, subpacked in a 2 mL cryopreservation tube, and stored at −80 °C for the high-throughput sequencing of microbial bacteria 16S V3-V4. Sequencing was completed by Beijing Biomarker Biotechnology Co., Ltd., Beijing, China. Total DNA was extracted from ileal and cecal digesta samples using the QIAamp Fast DNA Stool Mini Kit (Qiagen, Hilden, Germany); bacterial DNA was quantified using a Microvolume UV-Vis Spectrophotometer (NanoDrop™ One, Thermo Fisher Scientific, Waltham, MA, USA) and standardized at 5 ng/μL. PCR amplification was then performed; the conditions for the amplification were set up as follows: 3 min at 95 °C, 30 s at 95 °C (25 cycles), 30 s at 55 °C, 30 s at 72 °C, 5 min at 72 °C, and then held at 4 °C. The PCR products were cleaned, and the library was combined with the sequencing adapters and dual indices using the Nextera XT Index Kit (Illumina, San Diego, CA, USA). The PCR assay conditions were set up as follows: 3 min at 95 °C, 30 s at 95 °C (eight cycles), 30 s at 55 °C, 30 s at 72 °C, 5 min at 72 °C, and held at 4 °C. Next, for purification of the PCR products, AMPure XP beads were used. The libraries were quantified using a KAPA Illumina library quantification kit (KAPA Biosystems, Bellevue, WA, USA). Individual concentrations of the DNA libraries were calculated in nM based on the size of the DNA amplicons as determined by an Agilent 2100 Bioanalyzer (Agilent Technologies, Santa Clara, CA, USA). The original microbial sequencing data were imported into FLASH v1.2.7 software for overlap data splicing to obtain original tags. The splices were then removed using Trimmomatic v0.33 software to obtain clean tags from among low-quality fragments. Finally, the chimera was removed using UCHIME v4.2 software to obtain the final effective tags. Uparse v7.0.1001 was used to cluster the effective tags with 97% similarity, and representative operational taxonomic unit (OTU) sequences were screened. The SSUrRNA database was used to annotate the representative bacterial OTUs to obtain the bacterial composition at phylum and genus levels. Finally, the data were normalized through α diversity and β diversity analyses. The dilution curve and species distribution histogram were drawn using R software R-4.2.2.

### 2.4. Data Statistics and Analysis

Raw data, preliminarily processed using Excel 2023, were normalized by centering around the mean and homogenized to meet analysis requirements. One-way ANOVA was performed for data analysis by using SPSS 21.0 (IBM Corporation, New York, NY, USA). Differences between the experimental groups were compared using Duncan’s test. For microorganisms between the two groups, α statistical analysis of diversity and species distribution differences were completed using the SPSS 21.0 independent *t*-test module, and the significance level was set as *p* < 0.05.

## 3. Results

### 3.1. Effect of Jujube Powder on the Growth Performance of Broilers

Compared with the control group, the diet supplemented with 2%, 4%, 6%, 8%, and 10% jujube powder had no significant effect on the ADG, ADFI, and F/G of broilers in the starter and finisher periods (*p* > 0.05). However, the addition of 8% jujube powder significantly improved the ADG during days 0–42 (*p* < 0.05) (Table 3).

### 3.2. Effect of Jujube Powder on Apparent Nutrient Utilization in Broilers

Compared with the control group, the diet supplemented with 8% and 10% jujube powder significantly improved the apparent utilization of organic matter in the broilers (*p* < 0.05); it increased by 6.7% and 6.8%, respectively. Then, 10% jujube powder significantly improved the apparent metabolic energy of the broilers (*p* < 0.05), which was increased by 0.41%. The added jujube powder had no significant effect on the total tract apparent digestibility of dry matter and total tract nitrogen retention in the broilers (*p* > 0.05) (Table 4).

### 3.3. Effect of Jujube Powder on the Serum Immune Indices of Broilers

Compared with the control group, the diet supplemented with 4%, 6%, 8%, and 10% jujube powder significantly increased serum IgA, IgG, IgM, and sCD4 levels in the broilers. Only the 2% jujube powder had no significant effect (Table 5; *p* < 0.05).

### 3.4. Effect of Jujube Powder on the Serum Antioxidant Performance of Broilers

Compared with the control group, the diet supplemented with 2%, 4%, 6%, 8%, and 10% jujube powder significantly increased serum T-AOC and SOD contents in the broilers (Table 6; *p* < 0.05) and significantly reduced the serum MDA content (*p* < 0.05). In addition to the diet supplemented with 2% jujube powder, the diet supplemented with 4%, 6%, 8%, and 10% jujube powder significantly increased the serum GSH-Px content in the broilers (*p* < 0.05), of which the 10% diet led to the most improvement.

### 3.5. Effect of Jujube Powder on the Intestinal Microbiota of Broilers

#### 3.5.1. Ileal and Cecal Microbiota Sequencing and α Diversity Analysis

In total, 1,919,975 pairs of reads were obtained through the sequencing of 12 ileal and cecal samples from the control group and the jujube powder group (8%). A total of 1,602,596 clean tags were obtained after removing the splices, filtering out impurities, and removing chimeras, with an average of 133,549 clean tags per sample. Ileal and cecal microbiota α diversity results revealed that, the ACE and Chao 1 indices of the ileum and cecum (Table 7) were significantly higher in the jujube powder group than in the control group (*p* < 0.05). This indicated that the number of species in the ileum and cecum increased significantly after jujube powder was added to the diet.

#### 3.5.2. Effect of Jujube Powder on Microbial Species Composition in the Ileum and Cecum of Broilers

The top 10 species with a high relative abundance were screened according to the relative abundance ranking from high to low, and the results are shown through histograms (Figure 1 and Figure 2). The dominant phyla with a high relative abundance in the ileum of the control group and the jujube powder group were Firmicutes (60.72%, 73.13%), Cyanobacteria (19.08%, 8.35%), Proteobacteria (11.58%, 7.70%), and Bacteroidetes (7.34%, 9.93%) (Figure 1A). The dominant phyla with a high relative abundance in the cecum were Firmicutes (80.83%, 82.56%) and Bacteroidetes (13.48%, 12.52%) (Figure 2A). At the genus level, the abundance of *Enterococcus* (11.15%, 22.49%), *Lactobacillus* (8.45%, 18.82%), and *Candidatus_Arthromitus* (12.74%, 7.73%) (Figure 1B) was relatively higher in the control group and jujube powder group. The dominant genera with a high relative abundance in the cecum were *Faecalibacterium* (1.39%, 9.08%), *Ruminococcaceae_UCG-014* (7.36%, 7.54%), and *Alistipes* (5.68%, 6.43%) (Figure 2B).

#### 3.5.3. Effect of Jujube Powder on the Microbial Community Composition in the Ileum and Cecum of Broilers

Figure 3 presents the difference in the relative abundance at the level of ileal and cecal microbiota and genera within the top ten. At the phylum level, compared with the control, the relative abundance of Firmicutes and Bacteroidetes in the ileum significantly increased, whereas that of Proteobacteria significantly decreased (Figure 3A) (*p* < 0.05). The relative abundance of Bacteroidetes in the cecum decreased significantly, whereas that of Proteobacteria increased significantly (Figure 3C) (*p* < 0.05). At the genus level, compared with the control group, the diet supplemented with 8% jujube powder significantly increased the relative abundance of *Lactobacillus* and *Romboutsia* in the ileum, whereas it significantly decreased the relative abundance of *Enterobacter* (Figure 3B) (*p* < 0.05). The relative abundance of *Bacteroides* and *Faecalibacterium* in the cecum increased significantly (Figure 3D) (*p* < 0.05).

## 4. Discussion

Most studies have proven that jujube powder has various biological activities. It can provide nutrients for the body’s growth and development as well as regulate the body’s metabolism as a bioactive substance [33,34,35]. As a new feed material, jujube powder contains crude protein (7.43%), ether extract (7.22%), calcium (0.37%), phosphorus (0.18%), and gross energy (17.10 MJ/kg), which can effectively improve the performance of animal production. Diet supplemented with jujube powder can improve the growth performance of broilers [16]. Similar results have been observed in other animals [19,36,37] This study’s results are basically consistent with those of the aforementioned reports. The dietary addition of 8% jujube powder significantly improved the ADG of broilers during the whole period (0–42 days). This improvement is caused by the presence of active substances such as polysaccharides, vitamins, and flavonoids in jujube powder [38]. Of them, polysaccharides make jujube powder palatable and thus improve the feed intake of broilers. Second, flavonoids in jujube powder can promote insulin hormone release [39], which is beneficial for the digestive function of intestines in broilers [40]. These factors also affect the utilization efficiency of nutrients in broilers. The apparent nutrient utilization rate is among the most crucial indicators for measuring whether livestock and poultry can fully utilize the nutritional value of the diet. The apparent nutrient utilization rate is affected by factors such as the feeding level, feed processing technology, animal growth stage, and animal intestinal health. A diet supplemented with 15% jujube powder improved the apparent digestibility of the crude protein, neutral detergent fiber, and acid detergent fiber of goats [41]. Supplementation with 10% jujube powder significantly improved the apparent digestibility of crude protein, calcium, and phosphorus of layers [42]. This result is consistent with the results of the present study. The addition of 10% jujube powder significantly improved the apparent utilization rate of organic matter and apparent metabolic energy in broilers. When 2–8% jujube powder was added, the total tract nitrogen retention utilization rate of broilers improved [43]. However, this experiment found no improvement in the apparent digestibility of total tract nitrogen retention with the addition of jujube powder, which may be related to the variety, nutritional level, dietary type, and feeding environment of experimental animals.

IgA, IgG, and IgM are crucial indicators for measuring humoral immunity [44]. sCD4, which belongs to the immunoglobulin superfamily, is an auxiliary molecule of the T cell differentiation antigen and participates in the body’s cellular immunity. Polysaccharides in jujube powder can promote the maturation of cellular and humoral immune systems, significantly enhance the phagocytic capacity in low-immunity mice [45], significantly increase serum IgA, IgG, and IgM levels in broilers [46,47], and have the same effect on layers [48]. The results of the aforementioned studies are consistent with those of this experiment. The diet supplemented with 4–10% jujube powder significantly increased serum IgG, IgM, and IgA levels in broilers, and the sCD4 content exhibited a significant upward trend. The underlying mechanism may be that polysaccharides, cyclic nucleotides, phenolic compounds, and organic acids are abundant in jujube powder and can affect the expression of pro-inflammatory cytokines, reduce blood lipid activity, and improve immunity [34,38,49,50]. Some studies have suggested that the immune regulatory mechanism of jujube inhibits the phosphorylation of P-38 and JNK signal proteins. Jujube plays an anti-inflammatory role through the NF-kBand P38/JNK-MAPK signaling pathway, in which polysaccharides have a major role [35].

Many studies have reported that blood T-AOC, GSH-Px, SOD, and MDA contents can reflect the antioxidant capacity of the body [51,52,53]. GSH-Px and SOD are crucial peroxidase enzymes that can eliminate excessive free radicals and improve the antioxidant performance of the body [54,55]. Free radicals can increase the plasma lipoprotein level and stimulate cell membrane lipid oxidation [56]. Jujube powder can increase serum T-AOC and GSH-Px contents in laying hens and reduce the MDA content [57]. It can improve SOD, CAT, GPX, and T-AOC activities in the breast muscle of broilers [16] and the antioxidant activity of goat milk yogurt [58]. In this study, the diet supplemented with jujube powder also significantly increased the serum T-AOC and SOD contents and reduced the serum MDA content. The rich polyphenols in jujube powder can serve as a reductant [59] and have a strong antioxidant effect [60], which can reduce free radicals and transform them into a more stable state [61], followed by the improvement in the antioxidant capacity of broiler blood. Vitamin C and phenolic compounds also have a good ability to scavenge free radicals and prevent lipid oxidation [33,62,63]. Whether a single active substance or a combination of multiple active substances has an effect needs to be further explored. In a word, the active substances present in jujube powder can improve the body’s immunity and antioxidant capacity through some mechanisms (polyphenols in particular serve as reducing agents to reduce cell membrane lipid oxidation), which are worthy of recognition.

Intestinal microorganisms have a crucial regulatory role in host intestinal health and function [64]. Normally, intestinal microorganisms are in a dynamic balance, but factors such as diet, environment, temperature, and humidity can cause an imbalance of intestinal flora and affect the performance of animal production [65]. An increasing number of studies have proved that the bioactive substances of jujube powder can enrich intestinal probiotics, thereby improving the body’s immunity [66]. The underlying mechanism may be that water-soluble polysaccharides and insoluble fibers in jujube powder can enhance the immune function by influencing the intestinal microbiota αPD-L1 response rate and immune efficiency [67,68,69]. Similar to the previous results [70], 8% jujube powder added to the diet significantly increased the ACE and Chao 1 indices of broiler ileum and cecum. This indicates that the active substances in jujube powder and inulin can improve the intestinal microbial diversity and abundance of broilers. The increase in the number and diversity of microbes is conducive to the body’s health [71,72].

This experimental result is consistent with those of earlier studies [73,74]. Firmicutes, Proteobacteria, and Bacteroidetes are the dominant bacteria in the broiler ileum and cecum. Firmicutes are involved in short-chain fatty acid (SCFA) production, and SCFAs can synthesize cholesterol, reduce intestinal pH, and help enhance the immune response [75]. Moreover, Firmicutes are abundant in the intestines of many mammals. Butyric acid, produced during carbohydrate fermentation [76], can improve parenteral nutrition-induced intestinal mucosal immune injury [77] and regulate the proliferation of intestinal immune cells [78]. Proteobacteria are considered a marker of intestinal flora imbalance and can cause intestinal inflammation in some intestinal environments [79]. Bacteroidetes and *Lactobacillus* can regulate the intestinal microbial balance [80]. Bacteroidetes digest carbohydrates and use polysaccharides to produce acetic acid and propionic acid [81]. On the one hand, *Lactobacillus* can produce lactic acid, hydrogen peroxide, and bacteriocin to exert an antibacterial effect. On the other hand, it can compete with pathogenic bacteria for adsorption sites to resist their proliferation [82]. *Romboutsia* contributes to the health and development of the human intestine. The abundance of *Romboutsia* is high in the intestines of healthy people, whereas it is low in the intestines of patients with enteritis. Therefore, *Romboutsia* is closely related to host health [83]. *Enterobacter* is a common pathogenic bacterium that can produce adhesins and exotoxins, causing lesions at the host site and diarrhea [84]. A diet supplemented with the Achyranthes bidentata extract significantly increased the number of *Lactobacillus* in the broiler cecum, whereas it reduced the number of Escherichia coli and Salmonella. This is consistent with this study’s results [85]. This study found that 8% jujube powder added to the diet significantly increased the relative abundance of Firmicutes and Bacteroidetes at the lleal hilum level, whereas it significantly decreased the relative abundance of Proteobacteria. It also significantly increased the relative abundance of *Lactobacillus* and *Romboutsia* at the lleal level, whereas it significantly decreased the relative abundance of *Enterobacter*. This indicated that adding jujube powder significantly promoted beneficial bacteria and significantly inhibited harmful bacteria. *Faecalibacterium* is generally expressed in high abundance in patients with chronic enteritis and stage enteritis. *Faecalibacterium* is also associated with the occurrence of the broiler enterotoxicosis syndrome [86]. However, the 8% jujube powder added to the diet in this study significantly reduced the relative abundance of Bacteroidetes at the cecal hilum level, whereas it significantly increased the relative abundance of Proteobacteria. It also significantly increased the relative abundance of *Bacteroides* and *Faecalibacterium* at the cecal level. This result may depend on the test chicken variety, diet type, feeding environment, and sampling time, which needs to be further verified. The aforementioned report confirms our previous conjecture that active substances in jujube powder indeed improve the body’s overall immunity by affecting the intestinal microbiota.

## 5. Conclusions

This study found that adding 8% jujube powder to the diet significantly improved the ADG. Moreover, 8% and 10% jujube powder significantly increased the total tract of apparent organic matter digestibility, and 10% jujube powder significantly increased the apparent metabolic energy. The diet supplemented with jujube powder increased serum IgA, IgG, IgM, sCD4, T-AOC, SOD, and GSH-Px contents and reduced the serum MDA content. Moreover, the 8% jujube powder added to the diet significantly improved the intestinal microbiota of broilers. Jujube powder may enhance the immune level of broilers by improving their intestinal microbiota.

## Figures and Tables

**Figure 1 animals-13-03398-f001:**
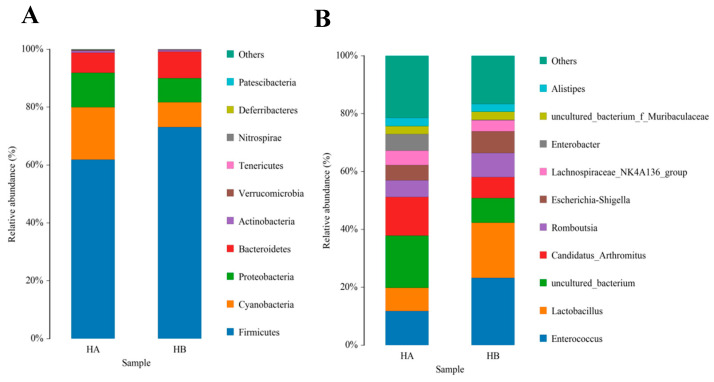
Effects of diet supplemented with 8% jujube powder on the ileal microbial composition of broilers. (**A**) is the ileal microbiological histogram at the phylum level; (**B**) is the ileal microbiological histogram at the genus level. HA is the control group; HB is the jujube powder addition group.

**Figure 2 animals-13-03398-f002:**
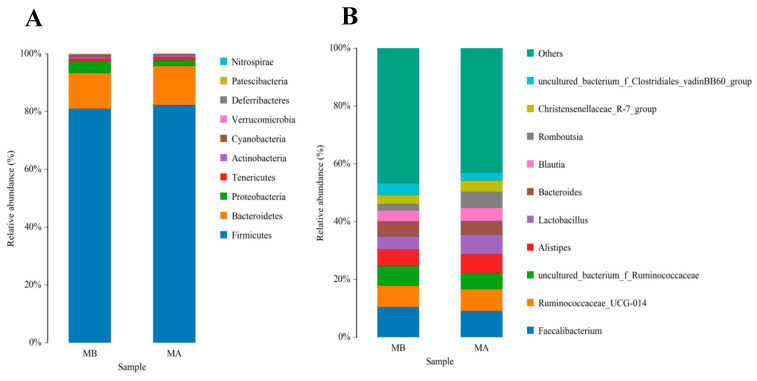
Effects of diet supplemented with jujube powder on the cecal microbial composition of broilers. (**A**) is the cecal microbiological histogram at the phylum level; (**B**) is the cecal microbiological histogram at the genus level. MA is the control group; MB is the jujube powder addition group.

**Figure 3 animals-13-03398-f003:**
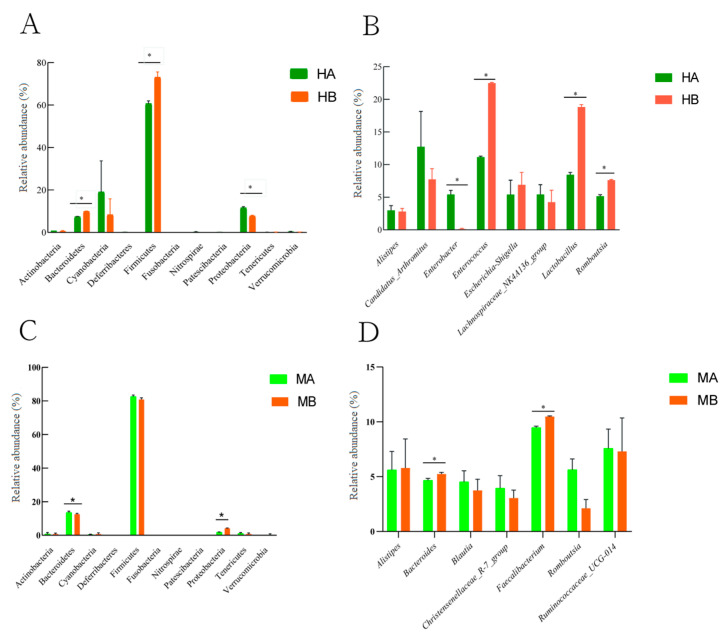
Differences in the microbial community composition between the ileum and cecum. (**A**) is the difference in the relative abundance of ileal microorganisms at the phylum level; (**B**) is the difference in the relative abundance of ileal microorganisms at the genus level; (**C**) is the difference in the relative abundance of cecal microorganisms at the phylum level; (**D**) is the difference in the relative abundance of cecal microorganisms at the genus level; * is the significant difference between the two groups (*p* < 0.05). HA and MA are the control groups; HB and MB are the jujube powder addition groups.

**Table 1 animals-13-03398-t001:** Nutrition content of jujube powder.

Items	Total Energy (MJ/kg)	CP (%)	EE (%)	CF (%)	Ash (%)	Ca (%)	P (%)
Jujube Powder	17.10	7.43	7.22	11.20	3.58	0.37	0.18

Note: CP: crude protein, EE: crude fat, CF: crude fiber, Ash: crude ash, Ca: calcium, P: phosphorus.

**Table 2 animals-13-03398-t002:** Composition and nutritional level of diets (air-dry basis, %).

Ingredients (%)	Starter						Finisher					
	0	2%	4%	6%	8%	10%	0	2%	4%	6%	8%	10%
Corn	64.16	61.39	58.76	55.86	53.36	50.76	62.90	60.10	57.40	54.60	51.90	49.10
Jujube powder	0.00	2.00	4.00	6.00	8.00	10.00	0	2.00	4.00	6.00	8.00	10.00
Soybean oil	2.00	2.50	2.90	3.40	3.60	3.90	3.50	4.00	4.50	4.80	5.30	6.00
Soybean meal, 46% CP	20.00	20.30	20.50	20.80	21.20	21.50	20.00	20.30	20.50	21.00	21.30	21.40
Cottonseed meal, 43% CP	3.00	3.00	3.00	3.00	3.00	3.00	3.00	3.00	3.00	3.00	3.00	3.00
Rapeseed meal, 35% CP	3.00	3.00	3.00	3.00	3.00	3.00	3.00	3.00	3.00	3.00	3.00	3.00
Distillers dried grains with solubles (DDGSs), 28% CP	3.00	3.00	3.00	3.00	3.00	3.00	3.00	3.00	3.00	3.00	3.00	3.00
Limestone	1.20	1.20	1.20	1.20	1.20	1.20	1.10	1.10	1.10	1.10	1.10	1.10
CaHPO_4_	1.70	1.70	1.70	1.70	1.70	1.70	1.20	1.20	1.20	1.20	1.20	1.20
Vitamin–mineral Premix ^1^	0.50	0.50	0.50	0.50	0.50	0.50	0.50	0.50	0.50	0.50	0.50	0.50
NaCl	0.40	0.40	0.40	0.40	0.40	0.40	0.40	0.40	0.40	0.40	0.40	0.40
DL-Methionine	0.15	0.15	0.15	0.15	0.15	0.15	0.20	0.20	0.20	0.20	0.20	0.20
L-lysine hydrochloride	0.50	0.50	0.50	0.50	0.50	0.50	0.40	0.40	0.40	0.40	0.30	0.40
NaHCO_3_	0.20	0.20	0.20	0.20	0.20	0.20	0.20	0.20	0.20	0.20	0.20	0.20
Choline chloride	0.20	0.20	0.20	0.20	0.20	0.20	0.20	0.20	0.20	0.20	0.20	0.20
TiO_2_	0	0	0	0	0	0	0.40	0.40	0.40	0.40	0.40	0.40
Total	100.0	100.0	100.0	100.0	100.0	100.0	100.0	100.0	100.0	100.0	100.0	100.0
Calculated nutrient content ^2^												
ME/(MJ/kg) ^3^	12.34	12.34	12.34	12.34	12.34	12.34	12.97	12.97	12.97	12.97	12.97	12.97
CP	20.00	20.00	20.00	20.00	20.00	20.00	18.00	18.00	18.00	18.00	18.00	18.00
Ca	1.00	1.00	1.00	1.00	1.00	1.00	0.80	0.80	0.80	0.80	0.80	0.80
AP ^3^	0.45	0.45	0.45	0.45	0.45	0.45	0.35	0.35	0.35	0.35	0.35	0.35
Lysine	1.15	1.15	1.15	1.15	1.15	1.15	1.05	1.05	1.05	1.05	1.05	1.05
Methionine	0.54	0.54	0.54	0.54	0.54	0.54	0.40	0.40	0.40	0.40	0.40	0.40

^1^ Premix is provided per kilogram of diet: VA, 15,000 IU; VD3, 5100 IU; VE, 25.5mg; VK, 2.5 mg; VB1, 3.6 mg; VB2, 6.9 mg; VB6, 4.5 mg; VB12, 0.036 mg; niacin, 66.3 mg; calcium pantothenate, 18 mg; biotin, 0.18 mg; folic acid, 1.4 mg; Mn (as manganese sulfate), 80 mg; Fe (as ferrous sulfate), 60 mg; Cu (as copper sulfate), 8 mg; Zn (as zinc sulfate), 90 mg; I (as potassium iodide), 1 mg; Se (as sodium selenite), 0.3 mg. ^2^ Nutrient contents were calculated using values from the NRC requirements. ^3^ ME: metabolizable energy, AP: available phosphorus.

**Table 3 animals-13-03398-t003:** Effects of jujube powder on the growth performance of broilers.

Jujube Powder Level	Starter (0–21)			Finisher (22–42)			Total (0–42)		
	ADFI (g/d)	ADG (g)	F/G	ADFI (g/d)	ADG (g)	F/G	ADFI (g/d)	ADG (g)	F/G
Control (0%)	41.41	27.11	1.51	132.57	70.10	1.88	86.9	48.6 ^b^	1.80
2%	41.91	27.95	1.49	133.38	71.24	1.87	87.7	49.6 ^ab^	1.77
4%	41.78	28.12	1.48	134.90	71.77	1.87	88.4	49.8 ^ab^	1.77
6%	41.85	27.75	1.51	134.76	72.61	1.85	88.5	50.0 ^ab^	1.76
8%	39.97	27.07	1.47	133.90	72.86	1.84	87.4	51.3 ^a^	1.73
10%	39.92	26.54	1.50	133.61	72.50	1.84	86.7	49.5 ^ab^	1.75
S.E.M.	0.48	0.39	0.01	0.70	0.41	0.01	0.58	0.65	0.01
*p* value	0.684	0.162	0.833	0.244	0.073	0.822	0.122	0.043	0.252

Note: S.E.M.: standard error for the sample mean. Different letters in the same row indicate a significant difference between the treatments (*p* < 0.05), and the same letter superscripts indicate no significant difference (*p* > 0.05).

**Table 4 animals-13-03398-t004:** Effects of jujube powder on the total tract apparent nutrient utilization digestibility or retention of broilers.

Jujube Powder Level	DM (%)	OM (%)	N (%)	AME (MJ/kg)
Control (0%)	79.2	77.2 ^b^	73.8	13.13 ^b^
2%	80.3	78.5 ^ab^	74.3	13.27 ^ab^
4%	79.5	79.6 ^ab^	74.6	13.38 ^ab^
6%	81.7	81.4 ^ab^	75.2	13.37 ^ab^
8%	81.5	83.9 ^a^	75.1	13.47 ^ab^
10%	82.6	84.0 ^a^	74.8	13.54 ^a^
S.E.M.	0.8	0.7	1.9	0.04
*p* value	0.864	0.039	0.765	0.045

Different letters in the same row indicate a significant difference between the treatments (*p* < 0.05), and the same letter superscripts indicate no significant difference (*p* > 0.05).

**Table 5 animals-13-03398-t005:** Effects of jujube powder on serum immune indices in broilers.

Jujube Powder Level	IgA (ng/mL)	IgG (ng/mL)	IgM (ng/mL)	sCD4 (ng/mL)
Control (0%)	295.70 ^d^	1853.42 ^e^	492.06 ^e^	68.57 ^e^
2%	320.92 ^d^	2060.12 ^de^	547.03 ^de^	75.50 ^de^
4%	455.02 ^a^	2273.15 ^cd^	729.99 ^b^	81.83 ^cd^
6%	405.49 ^ab^	3016.73 ^a^	829.50 ^a^	89.04 ^bc^
8%	376.25 ^abc^	2674.82 ^b^	667.70 ^bc^	95.63 ^ab^
10%	348.89 ^bc^	2468.15 ^bc^	612.48 ^cd^	107.40 ^a^
S.E.M.	10.9	63.7	17.38	2.2
*p* value	<0.01	<0.01	<0.01	<0.01

Different letters in the same row indicate a significant difference between the treatments (*p* < 0.05), and the same letter superscripts indicate no significant difference (*p* > 0.05).

**Table 6 animals-13-03398-t006:** Effects of jujube powder on the serum antioxidant properties of broilers.

Jujube Powder Level	T-AOC U/L	GSH-PX U/L	SOD U/L	MDA U/L
Control (0%)	9.06 ^e^	120.83 ^d^	227.89 ^e^	14.82 ^a^
2%	13.27 ^d^	146.65 ^cd^	349.34 ^d^	12.83 ^b^
4%	16.03 ^cd^	161.81 ^bc^	387.88 ^cd^	11.76 ^bc^
6%	18.45 ^bc^	177.24 ^ab^	430.38 ^bc^	10.66 ^cd^
8%	21.15 ^ab^	191.16 ^a^	471.75 ^ab^	9.58 ^de^
10%	24.08 ^a^	203.58 ^a^	513.66 ^a^	8.37 ^e^
S.E.M.	0.78	4.78	12.53	0.34
*p* value	<0.01	<0.01	<0.01	<0.01

Different letters in the same row indicate a significant difference between the treatments (*p* < 0.05), and the same letter superscripts indicate no significant difference (*p* > 0.05).

**Table 7 animals-13-03398-t007:** Effects of jujube powder supplementation on the ileal and cecal microbiota α diversity of broilers.

Items	Control	Jujube Powder Addition Group	S.E.M.	*p* Value
ileum microbiota				
ACE index	532.53	600.57	17.57	0.03
Chao 1 index	539.35	610.60	45.92	0.01
Simpson index	0.17	0.16	0.09	0.86
Shannon index	2.95	3.01	0.25	0.91
coverage	0.99	0.09	<0.01	1.00
cecal microbiota				
ACE index	498.77	519.27	7.69	0.02
Chao 1 index	507.55	540.62	8.33	0.03
Simpson index	0.03	0.03	<0.01	0.90
Shannon index	4.35	4.38	0.07	0.85
Coverage	0.99	0.99	<0.01	0.01

## Data Availability

The data presented in this study are available on request from the corresponding author.

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
