# Peer review of "Effects of Jujube Powder on Growth Performance, Blood Biochemical Indices, and Intestinal Microbiota of Broiler"

_animals, 2023, doi:10.3390/ani13213398_

Round 1
Reviewer 1 Report
Comments and Suggestions for Authors
Even though this paper provide likely useful data for using jujube powder in chicken feed, it is hard to say this idea would be successful, due to jujube collection, process and uncertain quality. I do not suggest this paper published in Animals.
Questions bellow should be concerned.
1、 what is the cost of jujube powder? Is it cheaper than corn?
2、 How many tons of jujube powder are available for feed each year in China?
3、 Fig1 dose not mean much, suggest to delete.
4、 The procession of jujube involved freeze, it is high cost.
5、 Nutrition of jujube used is not provided.
6、 How to control the nutritional value in different diets exactly the same?
7、 2.3.4: which treatment group you took 6 broilers for intestinal sample collection? Why choose the best growth performance group? How many samples were used in the following sequencing?
8、 Jujube powder used in this trail may not the same as samples used as report, thus, nutritional value from report may not be valuable.
9、 The idea of this paper is vague, replacement of corn, or as a functional component?
Reviewer 2 Report
Comments and Suggestions for Authors
Introduction:
Line 46: The sentence beginning with "Red jujube is abundant" should start a new paragraph.
Methods:
Line 78: "were then dried naturally." Were they air-dried or sun-dried?
Line 97: The dimensions provided should be placed after "cages" in line 98, not after "chickens" in line 97. Since a 3-layered cage was utilized, were all treatments represented on every layer, and was there blocking for cage layer?
Results:
Line 175: "However, the addition of..." needs revision for clarity. According to Table 2, 8% jujube powder only improved ADG during days 0–42 when compared to the Control group. The same applies to the result regarding 10% jujube on ME in lines 183-184.
Lines 189-191: The sentence "Compared with the control group..." is not entirely accurate, as statistical similarities are observed in Table 4. According to Table 4, 4% jujube powder IgG result, 10% jujube powder IgM result, and 4% jujube powder sCD4 result are exceptions to the summarized results presented in lines 189-191.
Lines 196-198: For the result on GSH-Px content, 2% and jujube powder had statistically similar results, as shown in Table 5. Revise the summarized result for clarity.
Line 206: "(Table 7)" should come after "cecum" in line 207.
Lines 230 and 235: "The same as below." These phrases are not needed and can be deleted.
Line 251: HA, HB, MA, and MB are not described in the Figure 4 description.
Discussion
Line 319: "through some mechanisms, which are worthy of recognition." Consider highlighting or speculating on some of these mechanisms here.
Conclusions:
Lines 371-374: In alignment with the presented results, please specify the exact inclusion level that recorded positive benefits in this section.
Comments on the Quality of English LanguageModerate editing of English language required
Reviewer 3 Report
Comments and Suggestions for Authors
Good and interesting work, evaluation of new feed ingredients is always welcome.
My only objection is that I think you should have provided a table with broiler weights (0, 2 and 42 days), because ADG is good data but depends on initial and final weights.
Reviewer 4 Report
Comments and Suggestions for Authors
Line 11 - in abstract and introduction it is unclear if you are targeting broilers or broiler breeders. you keep highlighting breeding broilers, but your study was conducted in broilers. If you want to highlight the importance of this product in broiler breeder reproduction, you should conduct a study in broiler breeders, not broilers.
in line 51, you say "jujube polysaccharides".....are polysaccharides found in jujube unique or different than those found in other fruits? if so, please go into further detail. if not, change the wording.
when you discuss jujube impact on cancer cells in line 61, can you specify what species was used in that study.
line 92 - why are you adding glucose to the drinking water? how does this impact the overall consumption of the birds. Birds that drink more water will receive a higher nutrient content. I imagine this will cause a lot more variation in your growth performance results. please justify.
Table 1 - you must include the analyzed content of the diet. not just the formulation.
In line 121, you say "good body condition". how do you define "good body condition"? what quantitative parameters are you using to determine body condition.
throughout the manuscript, when you mention an ELISA kit,, please cite the kit with a catalog number or other reference number as well as the manufacturer.
line 150 - why are you only selecting the largest birds from each group? how did you prevent bias in this selection? please justify.
section 2.3.4. please provide more information on how the sequencing was conducted. if you used a company to sequence, please provide company information and location. what methodology did they use.
Table 3 - why do you think the digestibility improvements in the 10% group did not result in improved growth performance. I would think that if OM and AME utilization is improved you should see improvements in growth. but you only see growth perfomance improvement in 8% group, not 10% group. please add this as a discussion point in the discussion section.
line 277- you say that jujube powder improves growth performance, but you only saw that in the 8% treatment group. please specify the inclusion level when you say it will improve growth performance. it will not always improve growth performance, according to your study results.
line 287- you say that apparent nutrient utilization is "the most intuitive test for analyzing broiler health". do you have a citation to support this claim? otherwise you need to remove it.
line 338 - how is jujube powder enriching probiotics?
line 343 - what do you mean by immune efficiency? how is the immune system improved in that paper?
Comments on the Quality of English Languagelines 9-10 should read "poultry diets" not "poultry diet"
in line 48 you say "fruit farmers and farmers". it would be better to specify poultry producers instead of just saying farmers.
line 69 - poultry breeding or poultry production?
line 302 - not sure what you mean by "indicating the T cell differentiation antigen". please clarify or reword.
there is an unnecessary parenthesis in line 314 and an unnecessary symbol in line 315
Round 2
Reviewer 1 Report
Comments and Suggestions for Authors
author have response clearly to most of my points. I belived this paper would add valuble information in feed resourse using in china.
Author Response
尊敬的审稿人:
非常感谢您对我们工作的认可,我们将在未来的研究中做出进一步的努力。
就这样。
此致